# Can We Harness Immune Responses to Improve Drug Treatment in Leishmaniasis?

**DOI:** 10.3390/microorganisms8071069

**Published:** 2020-07-17

**Authors:** Raphael Taiwo Aruleba, Katharine C. Carter, Frank Brombacher, Ramona Hurdayal

**Affiliations:** 1Department of Molecular and Cell Biology, University of Cape Town, Cape Town 7925, South Africa; arlrap001@myuct.ac.za; 2Strathclyde Institute of Pharmacy and Biomedical Sciences, University of Strathclyde, Glasgow G4 0NR, UK; 3International Centre for Genetic Engineering and Biotechnology, Cape Town Component, Cape Town 7925, South Africa; frank.brombacher@icgeb.org; 4Division of Immunology, Department of Pathology, Faculty of Health Sciences, Institute of Infectious Diseases and Molecular Medicine (IDM), South African Medical Research Council (SAMRC) on Immunology of Infectious Diseases, University of Cape Town, Cape Town 7925, South Africa; 5Faculty of Health Sciences, Wellcome Centre for Infectious Diseases Research in Africa, Institute of Infectious Diseases and Molecular Medicine (IDM), University of Cape Town, Cape Town 7925, South Africa

**Keywords:** leishmaniasis, chemotherapy, immunochemotherapy, host directed therapy, immunity

## Abstract

Leishmaniasis is a vector-borne parasitic disease that has been neglected in priority for control and eradication of malaria, tuberculosis, and HIV/AIDS. Collectively, over one seventh of the world’s population is at risk of being infected with 0.7–1.2 million new infections reported annually. Clinical manifestations range from self-healing cutaneous lesions to fatal visceral disease. The first anti-leishmanial drugs were introduced in the 1950′s and, despite several shortcomings, remain the mainstay for treatment. Regardless of this and the steady increase in infections over the years, particularly among populations of low economic status, research on leishmaniasis remains under funded. This review looks at the drugs currently in clinical use and how they interact with the host immune response. Employing chemoimmunotherapeutic approaches may be one viable alternative to improve the efficacy of novel/existing drugs and extend their lifespan in clinical use.

## 1. Introduction

Leishmaniasis, one of the World Health Organisation (WHO) top 20 neglected tropical diseases (NTDs), is caused by infection with kinetoplastid protozoans of the genus *Leishmania* (Table 1). These parasites are transmitted by phlebotomine sandflies of the genus *Phlebotomus* (in the Old World) or *Lutzomyia* (in the New World) during the process of taking a blood meal. Traditionally, infection is classified into three clinical types: self-healing cutaneous leishmaniasis (CL), disfiguring mucocutaneous leishmaniasis (MCL), and lethal visceral leishmaniasis (VL) [1], each type has diverse immunopathologies and levels of morbidity and mortality.

*Leishmania* has two main life-forms: the extracellular promastigote stage, which is present in the vector, the female sandfly, and the intracellular amastigote, which is the form present in the mammalian host. During sandfly feeding, deposited metacyclic promastigotes are endocytosed by phagocytes, where they differentiate into the amastigote forms, which survive the hostile oxidative environment of the host cell phagolysosome because of their robust superoxide dismutase and trypanothione redox systems [5,6]. Amastigotes multiply within infected cells, until cells burst releasing parasites that disseminate to species-specific sites of infection, ultimately causing the clinical symptoms associated with CL, MCL, or VL. Reservoir hosts such as dogs, are very important in the transmission of VL in endemic areas, and these hosts should be considered in clinical and veterinary *Leishmania* control programmes [7,8].

The WHO has identified leishmaniasis as a control priority; however, it is often overlooked in favour of research funding for HIV/AIDS, malaria, and tuberculosis. These diseases received 42.1% of the WHO health development research budget whilst NTDs only received 0.6%, which seems inadequate given their severity and associated mortality. This lack of investment may have a greater impact on the well-being of people in low-income countries, where up to five NTDs may be endemic [9,10]. NTDs such as leishmaniasis can also cause devastating lifestyle changes in terms of school attendance, intellectual abilities, labour productivity, and social stigma [11]. Between 700,000 to 1 million *Leishmania* infections are reported annually and one billion people residing in 98 countries are at risk of infection [12]. Leishmaniasis is estimated to cause 2.4 million disability-adjusted life years and 20,000–40,000 deaths/year [13,14]. Comorbidities are an added risk factor and VL is a common complication for human immunodeficiency virus (HIV) infected individuals. Co-infection increases susceptibility to VL by 23.2%, whilst VL elevates the progression of HIV to AIDs by 100−2320 × [15,16,17], as immune mechanisms required to control either disease is impaired [16]. Therefore, VL may pose a greater heath risk in *Leishmania*-endemic countries where HIV infected populations are present [18], although the total burden of this co-infection is underreported, partly due to the remoteness of affected areas. 

Although various chemotherapeutic agents have proven effective against leishmaniasis, the number of effective and affordable drugs are progressively declining due to the long treatment regimens, emergence of drug-resistant parasites, and treatment failure, making joint treatment protocols a preferred option. Ideally a vaccine to prevent infection is required but currently there is no clinically approved vaccine nor new drugs to combat drug resistant strains. Progress is impeded by a poor understanding of immune mechanisms that could elicit a long-lasting memory response by both B and T cells, which would allow clear criteria for predicting efficacy against the different *Leishmania* spp. [19,20]. On a more positive note, studies in our laboratory and that of collaborators have shown that host-protective immunity is achievable as an effective IFN-γ, T helper (Th)1-mediated immune response can aid in controlling the infection and enhancing treatment [21,22,23,24]. Therefore, with the new roadmap embarked by tropical disease research of the WHO towards prevention and control of leishmaniasis by 2030, and the Sustainable Development Goals (SDG1, SDG9, SDG11, SDG13, and SDG16) [25], a concerted effort is required from all stakeholders in order to move from bench to bed side. Platforms giving free access to characterized compounds e.g., OpnMe from Boehringer Ingelheim and Lilly’s Open Innovation Drug Discovery platform, is one option to facilitate novel drug development. 

Another way of improving drug treatment is to use chemoimmunotherapeutic approaches, where the host immune response is used to enhance the efficacy of a drug (Figure 1). Below, we discuss some of the drugs currently used in treatment of leishmaniasis, including drug resistance mechanisms and drug effects on host immune responses. We then further discuss types of immunotherapies and host-directed therapies that could be combined as immunochemotherapies towards a novel treatment regimen, whilst bearing in mind that any intervention must not select for genetic changes that enhance parasite survival within the host.

## 2. Benchmark Drugs for Leishmaniasis

Leishmaniasis was identified in the 1900′s and the first drugs were introduced in 1950. Seventy years later, we are still using these drugs [26,27]; hence, this infection remains a major public health problem in endemic areas where management relies solely on chemotherapy.

### 2.1. Pentavalent Antimonials

#### 2.1.1. Dosage and Side-Effects

Pentavalent antimonials (Sb^v^), sodium stibogluconate (SSG), or meglumine antimoniate, have been used extensively in the treatment of all clinical forms of leishmaniasis since the 1950′s [28,29]. Sb^v^ drugs are absorbed rapidly in the bloodstream of the host with a half-life of 2 h and a terminal mean half-life of 76 h when given intravenously [30]. These compounds have, however, faced several challenges in clinical settings. Antimony therapy requires daily parenteral administration for at least three weeks (20 mg Sb^v^/kg/day for 20–30 days) [31] and a registered medical practitioner for administration. Treatment can be associated with injection pain and toxic side-effects such as cardiotoxicity and renal failure [32]. This has influenced non-compliance leading to suboptimal dosing in resource-limited areas and probably caused emergence of drug resistant parasites [33]. Additionally, the long course of drug administration allows for cumulative effects, such as acute interstitial nephritis, myalgia, or death during or after treatment; hence, SSG therapy requires careful medical supervision [34,35]. These effects may lead to cessation of treatment before attaining curative levels.

#### 2.1.2. Mechanism of Action and Immune Response

The mode of action for antimonials is still poorly understood despite their long use for leishmaniasis. Several studies have demonstrated that Sb^v^ drugs are prodrugs, reduced to the more toxic and active anti-leishmanial Sb^III^ form, either in the parasite and/or the host cell [28,36,37]. Parasite-derived trypanothione reductase (TR) and a zinc-finger protein have been identified as potential molecular targets of Sb^III^ [31]. The *L. major* DNA hexamer binding protein (HEXBP), made up of nine CCHC zinc finger motifs, binds to the glycoprotein Gp63 on the promastigote surface, is involved in the process of DNA replication [38]. The TR system is crucial for maintaining cytosolic redox homeostasis and protects the parasite from toxic heavy metals, hence Sb^III^ stimulates a fast efflux of TR and thus reduces the thiol buffering capacity of the parasite [39]. Parallel investigations reported that adenine nucleoside and deoxynucleoside complexes inhibit purine transporters in *Leishmania* spp. and may act synergistically with other cytotoxic nucleoside products [40]. Antimonials are less effective in immunocompromised individuals [41] highlighting that immunocompetency is required for complete efficacy. Murine studies revealed that effectiveness of Sb^v^ relies on CD4^+^ and CD8^+^ T-cell subsets, the action of type-1 and type-2 cytokines (IL-2, IL-4, IL-12, IFN-γ and TNF) [42,43,44,45,46] and activation of reactive oxygen species and nitric oxide (NO) production in mouse macrophages [47]. Co-treatment of infected macrophages with exogenous IFN-γ and TNF-α significantly enhanced parasite killing and led to Sb^v^ accumulation, indicating that immunostimulation increases drug efficacy [43]. Studies in mice have demonstrated that SSG-mediated parasite clearance is organ-dependent, with the liver being more amenable to treatment compared with the spleen or bone marrow [48]. This can be partly explained by the pharmacokinetics profile of the drug, as very little SSG reaches the bone marrow after treatment [49]. VL is associated with granuloma formation in the liver and these immunological structures enhance the inherent activity of SSG [50].

#### 2.1.3. Antimony Resistance

Antimonials are no longer a first-line treatment for VL in some areas e.g., India, since emergence of parasite resistance limited its efficacy. Understanding how drug resistance has developed can help manage the usage of new drugs, so that clinical utility is not compromised [51,52,53,54]. The mechanisms responsible for Sb resistance in different species are still under-investigation as there are species-specific differences in efficacy. For example, *L. major* amastigotes in mouse macrophages are significantly less sensitive to SSG than *L. donovani* amastigotes [55], and *L. mexicana* is less sensitive to Sb^v^ than *L. braziliensis* [55]. There are now several studies published in this area [56,57,58] and important for resistance are parasite proteins involved with drug efflux e.g., LABCG2 [59] and aquaglyceroporin 1 [60].

### 2.2. Amphotericin B

#### 2.2.1. Dosage and Side-Effects

Amphotericin B (AmB) was introduced in the 1970′s for the management of systemic fungal infections [61]. It is administered intravenously due to it poor absorption by the gastrointestinal tract. For leishmaniasis, an infusion of 1 mg/kg, given daily for 20 days or 15 infusions of the same dose over 30 days, had a 95% cure rate [62,63,64]. However, infusion-related side effects e.g., nephrotoxicity, exacerbated by a long-treatment regimen, compromised its efficacy [43,65,66]. The induced nephrotoxicity affects blood vessels and epithelial cells leading to a decrease in glomerular filtration and tubular dysfunction, respectively [65]. Treatment with lipid formulations of AmB e.g., liposomal AmB (AmBisome^®^), AmB lipid complex (Abelcet^®^), and AmB cholesterol dispersion (Amphocil^®^) reduced drug toxicity and allowed shorter treatment regimens. Cure rates of 90–100% have been achieved in VL using a 5–7-day treatment regimen [62,67,68], and treatment at higher doses e.g., 10 mg/kg AmB. The liposomal carrier system improved the drug’s pharmacokinetic profile such that more of the drug was targeted to parasitized host cells in tissues rather than the kidneys [69]. Initially, clinical use was limited due to the high cost of liposomal AmB, but this was reduced from $200 per 50-mg vial to $20 per vial, making its use viable in resource-limited developing countries [70,71]. Response rates to lipid formulations of AmB varies, and hence higher doses are administered in Brazil, Eastern Africa, and the Mediterranean to induce cure [63], which may be parasite/host related. Despite excellent data on VL, there are scanty investigations on the role of liposomal AmB on CL and MCL and most are retrospective studies [72,73,74]. For HIV-*Leishmania* co-infected patients, liposomal AmB is the first line drug, where a cumulative dose of 40 mg/kg is recommended by the WHO [75,76].

#### 2.2.2. Mechanism of Action and Immune Response

The activity of AmB against *Leishmania* is linked to its ability to bind ergosterol in the parasite’s cell membrane, inducing pore formation and lysis of the parasite [77,78]. Indeed, ketoconazole-induced depletion of ergosterol in *L. mexicana* promastigotes reduced the lytic effect of AmB [79]. Studies indicate that AmB activity is not reliant on an intact immune system as AmB was equally active in clearing parasites in *L. donovani*-infected euthymic and nude BALB/c mice lacking T-cells [45]. However, AmB has been shown to have immunomodulatory effects as it can accumulate NO (nitric oxide) and ROS (reactive oxygen species), stimulate surface receptors and production of multiple immune mediators (cytokines, chemokines, and prostaglandins) [77]. Accordingly, joint treatment of AmB and immunological mediators (IL-12 or anti-IL-10 receptor) to boost host Th1 responses significantly enhanced the activity of AmB against *L. donovani* [80]. This may explain why HIV-*Leishmania* coinfection is associated with an increased risk of drug unresponsiveness, relapse and therapy-related mortality [81].

#### 2.2.3. AmB Resistance

Resistance to AmB was experimentally induced in *L. donovani* strains and clinically detected in 2012 [82], associated with an alteration in ATP-binding cassette transporters, membrane composition, ROS scavenging and upregulated thiol metabolic pathways [82]. Omic studies have identified a mutated sterol 14α-demethylase as an AmB resistance marker, which triggers a change in sterol metabolism [83]. Having a resistance marker to map emergence of drug resistance foci and knowing the exact mechanism(s) a drug uses is essential for developing a management plan to ensure that a drug remains in clinical practice for as long as possible.

### 2.3. Paromomycin

#### 2.3.1. Dosage and Side-Effects

Paromomycin (PR) is an inexpensive antibiotic with broad-spectrum activity against bacteria [84] and intestinal parasites [85,86]. PR is used in the treatment of CL, via topical and parenteral administration, and VL, via parenteral administration only [87,88,89]. At a dose of 15 mg/kg (11 mg base) for 21 days, PR gives a 95% cure rate in VL patients [54]. Nephrotoxicity, vestibular, and cochlear are associated side-effects; however, there are as yet no reported cases of this in VL therapy [90]. Ototoxicity has been reported in 2% of 442 patients on PR, with the most frequent side-effect being injection site pain [91]. Topical formulations of PR are the most commonly used since the hallmark of CL is skin lesions and it is therapeutic against both Old World leishmaniasis (OW) and New World leishmaniasis (NW) CL [92,93]. Evidence suggests that using the drug is beneficial but there are contradicting data/results [92], which probably relate to poor drug penetrance, which may be related to CL lesion pathology.

#### 2.3.2. Mechanism of Action and Immune Response

Mechanistically, PR alters the viscosity of the lipid-bilayer, respiratory chain, essential mitochondrial activities and lipid metabolism of the parasite [94,95], but the main target is thought to be inhibition of protein production. Studies have shown that PR targets the decoding A site of the small subunit of ribosomes within the cytoplasm rather than the mitochondria, where it causes misreading and translation inhibition [96]. There are no studies showing that PR has a direct effect on host immune responses.

#### 2.3.3. PR Resistance

Paromomycin resistant strains of *Leishmania* promastigotes or amastigotes can be induced experimentally [97,98]. A Nepalese strain of *L. donovani* had a high natural resistance to PR and increased resistance to NO was noted. PR resistance was shown to be associated with lipidomic and metabolomic strain-specific changes [99]. A recent study using chemically induced mutagenesis identified a putative calcium dependent protein kinase (LinJ.33.1810) as a potential resistance marker [56].

### 2.4. Miltefosine

#### 2.4.1. Dosage and Side-Effects

Miltefosine (MIL), an alkyl phosphocholine originally used for cancer treatment, has been repurposed as the only oral drug for the treatment of VL and CL. MIL was approved for use in India in 2002, for kala-azar, after 50 mg and 100 mg doses achieved a 94% cure rate in a randomized, open-label, phase 3 clinical trial [100,101]. This was supported by a phase 4 study in an outpatient setting in India [102]. Regardless, in eastern Africa, MIL did not achieve up to 90% cure rate; this regional difference could be attributed to genetic diversity and drug susceptibility in *Leishmania* strains [103]. In Ethiopia, MIL is safer but less effective than Sb in HIV-*Leishmania* infected patients but is not inferior to standard Sb therapy in non-HIV infected patients [104]. Hence, treatment of VL in eastern Africa remains a big problem as both AmBisome and MIL are suboptimal. MIL is effective against CL, but results vary between species. For instance, in Columbia, MIL achieved a per-protocol cure-rate of 91% for CL caused by *L. panamensis,* whereas in Guatemala, only a 53% per-protocol cure-rate was obtained for *L. braziliensis* and *L. mexicana* [105], well below the cure-rate achieved by antimony. However, MIL (75% cure rate) proved to be more efficacious than antimony (53% cure rate) in Brazilians with CL caused by *L. braziliensis*, signifying intrinsic sensitivity of this *Leishmania* spp. in different regions [106]. There is a high risk of relapse associated with MIL treatment, probably because host immunity contributes to drug efficacy [107]. Altogether, MIL is the main CL recommended leishmanicide because it is the only orally administered drug and is tolerable with low-grade side-effects.

#### 2.4.2. Mechanism of Action and Immune Response

As with most of the anti-leishmanial drugs, the mechanism of action of MIL is not well-defined. It is believed that the drug disrupts lipid metabolism, mitochondrial dysfunction, and induces apoptosis [108]. This may be partly caused by the drug’s ability to disrupt calcium homeostasis, resulting in an increase in intracellular calcium accumulation [109]. However, the action of MIL on host immune responses may also contribute to its antiparasitic efficacy. For instance, MIL treatment increased IFN-γ, TNF-α, and IL-12 production by *Leishmania*-infected mice and patients [110]. Additionally, treatment of macrophages with MIL increases macrophages phagocytosis and could significantly increase NO production of infected but not uninfected macrophages [111]. Finally, MIL increases macrophage expression of IFN-γ and IL-12 production via enhanced CD40 expression or CD40-induced p38MAPK phosphorylation [112]. Overall, these studies indicate that MIL would favour the development of a protective Th1 immune response.

#### 2.4.3. MIL Resistance

Studies have revealed that *Leishmania* resistance to MIL is possible and is associated with decreased drug accumulation, caused by overexpression of an ATP-binding cassette transporter P-glycoprotein, or decreased uptake through the inactivation of the MIL transporter LdMT and its beta subunit LdRos3 [113]. However, MIL resistance was associated with strain-specific changes and deletion of the LdMT gene only occurred in one strain. Moreover, MIL resistance is linked with metabolites associated with the Kennedy pathway in MIL resistant parasites [114]. A reduction in treatment efficacy was reported in India in 2012 [115] and Nepal in 2013, but analysis of drug susceptibility did not relate relapse with increased parasite drug resistance [33]. However, by 2017 isolates with enhanced resistance to MIL were identified [116], indicating that the only oral drug may be compromised if its use is not controlled.

### 2.5. Host-Directed Therapeutics

Host-directed therapies (HDTs) focus on improving the ability of the host to defend itself against infectious (and non-infectious) agents [117]. Pathogen survival can be negated by modulating redundant host molecules or immune pathways that are essential for invasion, survival/replication, or clearance of the parasite [118]. HDTs may be implemented as a sole treatment approach targeting immunological pathways as an immunotherapy (Figure 2) or in conjunction with established drug treatments as chemoimmunotherapy [119]. Hierarchically, chemoimmunotherapy is superior as it offers a synergistic approach, with activation of the immune system combined with a direct parasiticidal action of drugs against the pathogen. Importantly, it has the potential to mitigate drug resistance because resistance to host molecules, while not completely impossible, is less frequent and is significantly more complex than antimicrobial resistance, which may be a single point mutation in the pathogen [120].

*Leishmania* infection involves a complex interplay between the host and parasite; hence, drug efficacy could be enhanced or compromised by the host’s immune response. Studies have shown that Th1-related cytokines can be used to control *Leishmania* infection and an additive effect is achieved when co-administered with chemotherapy. Accordingly, treatment with IL-2 reduced *L. donovani* parasite burdens by 50% [121], IL-12 treatment reduced parasite burdens by 47%, and IFN-γ treatment by 40% [122]. Treatment with IFN-γ alone was only marginally effective in VL patients [123] but was more effective in combination with meglumine antimoniate, which significantly improved the clinical condition for 14 out of 17 patients, who had manageable side effects such as fever, fatigue, myalgia, and headache [124]. Similarly, combination therapy with IL-12 or an anti-CD40 antibody to manipulate costimulatory pathways, increased the efficacy of a suboptimal dose of AmB against *L. donovani* [80]. Anti-CD40 therapy has been tested in cancer patients and has been shown to cause a dose-related upregulation of costimulatory molecules after treatment [125]. Joint treatment with anti-CD40 antibody in combination with cisplatin and pemetrexed resulted in long term survival of three out of 15 patients, but a “cytokine storm” reaction occurred in all patients [126]. This highlights the need to ensure that the immunostimulator does not induce adverse side-effects. Furthermore, cytotoxic T lymphocyte Ag-4 (CTLA-4), a negative regulator of T cell activation [127], is a promising immunomodulatory target. Inhibiting CTLA-4 enhanced the frequency of IFN-γ and IL-4-producing cells in both spleen and liver of *L. donovani*-infected mice and augmented the development of hepatic granulomas (Figure 2), altogether enhancing host resistance to infection [128]. Co-administration of antimonial with anti-CD40 and anti-CTLA-4 [129], or an OX40L fusion protein and anti-CTLA4 [130] synergistically enhanced leishmanicidal activity and boosted granuloma maturation in mice. Evidently, costimulation-based immunochemotherapies are promising and warrant clinical investigations in *Leishmania*-endemic regions are needed.

Dendritic cells (DCs) prime CD4^+^ and CD8^+^ T cell responses forming a critical link between innate and adaptive immunity. Studies have shown that early IL-4 instruction of DCs to produce IL-12 via decrease of IL-10 may mediate a beneficial Th1 response [130,131,132,133]. This is interesting because IL-4 is generally recognized as a canonical Th2 cytokine, yet paradoxically it can instruct Th1 immunity [134,135], suggesting that DCs may represent a checkpoint in chemo-immunomodulation. Furthermore, vaccination with soluble *Leishmania donovani* Ag-pulsed DCs combined with Sb treatment resulted in sterile immunity in *L. donovani* infected mice [136]; thereby, substantiating the use of immune cells as alternate immunochemotherapeutics. In fact, any compound capable of enhancing the phagocytic ability and oxidative burst of phagocytic cells (macrophages, neutrophils, or DCs) would constitute ideal targets for immunochemotherapy in *Leishmania*. Apart from modulating DC presentation, IL-10 also downregulates Th1 responses, macrophage activation, and pathways necessary for reactive nitrogen intermediates [137]. Accordingly, the demonstration that anti-IL-10R was remarkably active by itself to limit IL-10 functionality, inducing a curing phenotype, and acted synergistically with antimony, substantiates its potential for immunochemotherapy [138,139,140].

There are other types of HDTs that could be combined with chemotherapy as immunochemotherapies. For example, it has been shown that *Leishmania* relies on host lipids to evade and manipulate the immune system [141], which can be alleviated by treatment with simvastatin, a cholesterol lowering drug that targets hydroxy-3-methylglutaryl coenzyme A reductase in cholesterol biosynthesis. Simvastatin treatment enhanced host specific Th1/Type 1 immunity by increasing leishmanicidal activities of host macrophages (Figure 2) and accelerated tissue repair of *L. major* lesions in a murine model [142]. Differentiation from promastigote to amastigote requires higher cholesterol levels [143] and within the parasitophorous vacuole in the mammalian host, simvastatin treatment can restrict this need since amastigote cholesterol originates from host membrane cholesterol [144]. This suggests that topical and systemic simvastatin could be used in conjunction with current anti-*Leishmania* drugs to enhance host-protective immunity and accelerate healing of lesions [144]. Likewise, formulations with stearylamine (SA)-bearing phosphatidylcholine liposomes together with paromomycin [145], sodium-antimony gluconate [146], or AmB [145] promote a beneficial Th1 response with concomitant downregulation of IL-10.

Impaired Th1 responses and killing-effector functions in the host during leishmaniasis is linked to downregulation of the mitogen-activated protein kinase (MAPK)/NF-κB and JAK/STAT signaling pathways [147] via a decrease in macrophage-derived TNF-α, NO, iNOS, IFN-γ, and IL-12 [148]. Fucoidan, a natural polysaccharide isolated from brown algae, suppresses this response by activating p38 and ERK1/2 related to the MAPK/NF-κB pathway to increase levels of IL-12, IFN-γ, iNOS, and TNF-α (Figure 2) [149,150]. As killing of intracellular *Leishmania* is critically dependent on IL-12, IFN-γ, and TNF-α, which enhances iNOS-derived NO production via classically activated macrophages, Fucoidan could provide a novel immunochemotherapeutic approach. Another target with similar effect is cystatin, a natural cysteine protease inhibitor, which activates ERK1/2, NF-κB, and JAK/STAT pathways for a curative effect to experimental VL [151]. Since MIL restores IFN-γ responsiveness together with IFN-γ-induced STAT-1 and p38MAP kinase phosphorylation [112], combination of Fucoidan or cystatin with MIL warrants further investigation as immunochemotherapies.

## 3. Conclusions

Leishmaniasis is a severely neglected disease despite the immense suffering it places on the host, especially in regions of economic instability. In turn, this leads to continued poverty and retarded socioeconomic development. Unfortunately, our arsenal of chemotherapeutic agents, in use since the 1950s, do not achieve a sterile cure and are generally toxic. In view of this, the WHO has strongly recommended more research into new drugs for leishmaniasis to reach a zero-death rate by 2030, in line with the SDG goals. Such strategies should aim to potentiate chemotherapeutic agents with specific immunomodulators or identify novel compounds that are both parasiticidal and immunomodulatory. There are limited studies on immunological treatments to boost the efficacy of drugs; however, researchers could use therapeutics identified in other studies (e.g., cancer treatment), as co-treatments and ensure they do not induce adverse side-effects e.g., cytokine storm [152]. Most drug discovery programs focus on cytotoxicity and efficacy of compounds, and overlook their dependence on innate and adaptive immunity, which could be harnessed to improve efficacy. This is rather unfortunate as different aspects of the host immune response such as regulation of cytokines, costimulatory pathways, macrophage activation, and cell signaling pathways, are clearly promising targets for immunochemotherapy. There are few clinical studies in this area, and this probably reflects the high cost of cytokine-, cell-, or antibody-based therapeutics for a NTD, but essentially, more are needed. To this end, drug-discovery platforms and pharmacological studies need to include immunomodulation in their analysis, so that their experimental models “mimic” the type of immunosuppression that can occur in the clinic.

## Figures and Tables

**Figure 1 microorganisms-08-01069-f001:**
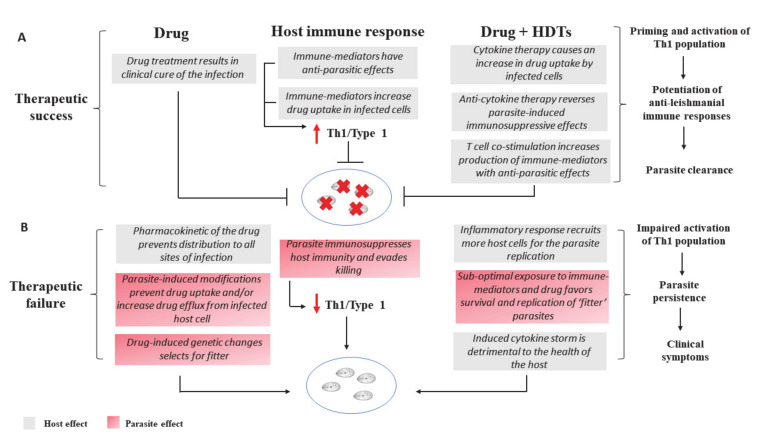
Proposed outcome of management of leishmaniasis using chemoimmunotherapeutic approach. (**A**): Therapeutic success can be achieved via three possible routes. Administration of drugs achieves clinical cure which inhibits parasite replication within host cells. In parallel, the host immune response could be activated by immune-mediators and directly or indirectly disrupt parasite growth or magnify drug uptake in host cells via upregulation of a specific Th1 response. This drives the activation of macrophages. Since both of these approaches are associated with different levels of success, a third approach is to combine the drug with a host-directed therapeutic or therapeutics in a chemo-immunotherapeutic approach, where host immune responses are targeted in conjunction with the drug, overall, potentiating an enhanced Th1 response and leishmanicidal effect. (**B**): Therapeutic failure occurs if an antileishmanial drug does not reach the appropriate sites of infection. Alternatively, this occurs if the drug induces genetic changes by selecting for “fitter” parasites that are resistant to the drug and/or oxidative stress or if intracellular parasites induce modifications to alter drug uptake and/or efflux from infected host cells, both of which render parasites less responsive to treatment. In parallel, *Leishmania* parasites are immunosuppressive; hijacking host immunity and impairing Th1 differentiation by inducing factors that enable disease by reducing the efficacy of the host immune responses such as, downregulating macrophage-derived nitric oxide whilst concomitantly, enhancing IL-10 to deactivate macrophage killing effector functions. An expert understanding of the host immune response is needed to implement the third approach as a chemoimmunotherapeutic regimen could exacerbate inflammatory responses. Enhanced inflammation could enhance recruitment of host-cells for parasite replication, select for fitter parasites, which are drug-resistant and unresponsive to host antileishmanicidal products, or induce a cytokine storm that is ultimately detrimental to the host.

**Figure 2 microorganisms-08-01069-f002:**
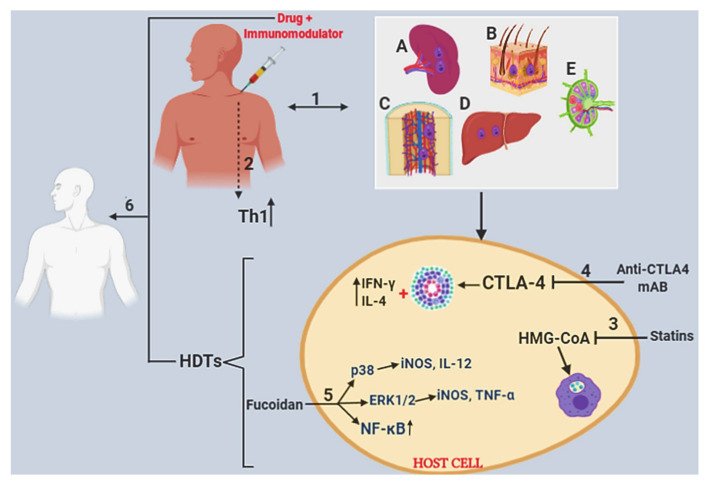
Host-directed therapies and immunochemotherapies and the protective roles they play in the mammalian host. (**1**) Various organs (**A**, infected spleen; **B**, infected skin; **C**, infected bone marrow; **D**, infected liver; and **E**, infected lymph node) are infected with *Leishmania*, the site depending on the infecting spp. (**2**) Administration of a drug in combination with an immunomodulator e.g., cytokines, co-stimulatory/inhibitory molecules, immune cells, which increase the antileishmanial Th1 population and boosts subsequent parasite clearance (3–5), represents a host-directed therapeutic drug target. (**3**) The administration of statins (an inhibitor of hydroxy-3-methylglutaryl coenzyme A reductase) enhanced host macrophage phagosome and killing effector function by lowering host cholesterol. (**4**) Inhibiting the cytotoxic T lymphocyte Ag-4 led to increased granuloma formation and IFN-γ and IL-4 production by the liver and spleen, which induced host-protective immunity. (**5**) Fucoidan activated p38 and ERK1/2 pathways leading to increased iNOS production, whereby p38 increased IL-12 levels and ERK1/2 enhanced TNF-α and NF-κB signal transduction for host-protective immunity. (**6**) Combination of drugs and HDTs leads to parasite clearance. Figure was created in BioRender.com.

**Table 1 microorganisms-08-01069-t001:** Different *Leishmania* species and their geographical location and pathologies.

Leishmania Spp	Vector	OW/NW	Location	Type of Disease
*Leishmania major*	*Phlebotomus papatasi* *P. ansari* *P. caucasicus* *P. bergeroti* *P. sergenti*	OW	Middle East, North Africa, Asia	CL
*Leishmania donovani*	*P. argentipes* *P. martini* *P. chinensis* *P. orientalis* *P. alexandri* *P. celiae*	OW	East Africa, India subcontinent	VL
*Leishmania infantum*	*P. alexandri* *P. ariasi* *P. langeroni* *P. longicuspis* *Lutzomyia migonei* *L. longipalpis* *L. cortelezzii*	OW	Central and South America, Mediterranean regions, Asia	VL
*Leishmania siamensis*	*Sergentomyia (Neophlebotomus) gemmea*	NW	Thailand, USA, Central and Western Europe	DCL/VL
*Leishmania braziliensis*	*L. longipalpis* *L. ayrozai* *L. lichyi* *Warileya rotundipennis* *L. shawi* *L. whitmani* *L. (Pintomyia) fischeri* *L. wellcomei*	NW	South America	MCL
*Leishmania mexicana*	*L. gomezi* *L. trapidoi* *L. anthophora* *L. ovallesi* *L. diabolica*	NW	North and South America	CL
*Leishmania amazonensis*	*L. evansi* *L. diabolica* *L. longipalpis* *L. (Nyssomyia) flaviscutellata*	NW	Amanzonas	CL/MCL
*Leishmania venezuelensis*	*L. olmeca* *L. lichyi* *L. rangeliana*	NW	Western Venezuela	CL
*Leishmania aethiopica*	*P. longipes* *P. sergenti*	OW	East Africa	CL/MCL
*Leishmania* tropica	*P. guggisbergi* *P. arabicus* *P. chabaudi*	OW	Middle East, North Africa, Asia	CL
*Leishmania panamensis*	*L. panamensis* *L. gomezi*	NW	Panama, Colombia	CL
*Leishmania equatoriensis*	*L. hartmanni*	NW	Ecuador	CL/MCL
*Leishmania peruviana*	*L. verrucarum* *L. peruensis*	NW	Peru	CL
*Leishmania* pifanoi	*L. flaviscutellata*	NW	Venezuela	CL/DCL
*Leishmania* colombiensis	*L. gomezi* *L hartmanni* *L. panamensis*	NW	Santander, Columbia	CL
*Leishmania guyanensis*	*L. anduzei* *L. umbratilis* *L. shawi*	NW	Brazil	CL
*Leishmania naiffi*	*L. ayrozai* *L. squamiventris*	NW	Brazil	CL
*Leishmania lainsoni*	*L. ubiquitalis* *L angelsi*	NW	Amanzonas, Ecuador, Peru, Bolivia	CL
*Leishmania enriettii*	*L. gomezi* *L. correalimai*	NW	Ghana, Florida, Central Europe	CL

OW: Old World leishmaniasis; NW: New World leishmaniasis; and DCL: Diffuse cutaneous leishmaniasis [2,3,4].

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
