# Peer review of "Can We Harness Immune Responses to Improve Drug Treatment in Leishmaniasis?"

_microorganisms, 2020, doi:10.3390/microorganisms8071069_

Round 1

Reviewer 1 Report

Dear Author/s,

The submitted review manuscript “microorganism-859560” is well written and structured draft which emphasizes the Leishmania mediated immune responses and also discussed the past/present available drugs, their treatments and resistant. Paper has also highlighted some loop-wholes and gap in research towards novelty and interventions of the Leishmania research. Author/s has illustrated and encapsulated all major factors governing the Leishmania epidemiology, drug resistance, immune response, etc. However, I would recommend taking care of the following minor points -

The manuscript “Can we harness immune responses to improve drug treatment?” is submitted under special issue “Treatment of Leishmaniasis” belongs to the section "Parasitology" of journal Microorganisms which is known among author/s, editor and reviewer/s. I hope this issue will be available online open access manuscript without highlighting the section etc; hence as a reader point of view, title of the manuscript should have the term “Leishmaniasis” that is what whole manuscript is talking about.

There are numerous neglected tropical diseases (NTDs); Leishmaniasis (Black fever) is one of them. Comparison of diseases (that mainly affect the small population of world versus other disease which are cosmopolitan) is seems like invalid in terms of severity and mortality. Evidently, most of the discussed and cited researches in the manuscript is also belongs to L. major and L. donovani parasites which is an African and Asian species. As far as progress of ongoing Leishmania research and funding is concerned it is the issue of concerned research fraternity.

In table 1 author/s has mentioned New World/Old World (NW/ OW) Leishmania species which is also understandable from their location. It will be more informative if you would mention here their vector phlebotomine sand flies species name, conditionally if any report is available in this perspective with the fact that this tropical disease transmitted or carried by only particular species of vector. This will also add-in information of host-pathogens interactions and valued the paper significantly.

In figure 1 legend, HDT should be explained as Host-directed therapeutics somewhere. Do not use "neglected NTD" together (Line 357). Improve the quality of figures if possible. These kinds of small mistakes should be taken care of author/s end.

Overall, paper is demanding some proofreading and doing the self-assessments/minor corrections including the cited references, spelling and English language. In addition to that, provide the justified answers of the above comments or do the appropriate/necessary amendments in the manuscript.

I appreciate the efforts made by authors to compile the ongoing research to treat the Leishmaniasis and its drug resistance, demanding such reports of other overwhelming NTDs in future too.

All the very best.

Author Response

We would like to thank the reviewer for his/her kind words and thank them for their time.  In response to the comments we have made the following changes to the manuscript.

  1. Title: we agree with the author and apologise for this oversight. 

The original title ‘Can we harness immune responses to improve drug treatment’ amended to ‘Can we harness immune responses to improve drug treatment of leishmaniasis?’

  1. We agree that leishmaniasis is not the only neglected topical diseases and that the problem is where funds are directed on the basis of severity and mortality.

The text that read:

‘The WHO has identified leishmaniasis as a control priority, however, it is often overlooked in favour of research funding for HIV/AIDS, malaria and tuberculosis. Notably, 42.1% of health development budget was used for these conditions whereas only 0.6% of the budget was used for NTDs yet low-income countries are affected by at least five NTDs accounting for 56 million disability-adjusted life years (DALYs) [7,8]. Mostly, leishmaniasis afflict poor populations, causing devastating lifestyle changes in terms of school attendance, intellectual abilities, labour productivity and social stigma’

has been amended to:

‘The WHO has identified leishmaniasis as a control priority, however, it is often overlooked in favour of research funding for HIV/AIDS, malaria and tuberculosis. These diseases received 42.1% of the WHO health development research budget whilst NTDs only received 0.6%, which seems inadequate given their severity and associated mortality. This lack of investment may have a greater impact on the well-being of people in  low-income countries, where up to five NTDs may be endemic [9,10].  NTDs such as leishmaniasis can also cause devastating lifestyle changes in terms of school attendance, intellectual abilities, labour productivity and social stigma’

Table 1 – we have amended this table to show the vectors responsible for transmission of each species.

  1. The legend in Figure 1 has been amended to take into account the comment of HDT.

The text that read:  

‘Since both of these approaches are associated with different levels of success, a third approach is to combine the drug with a host-directed therapy, where host immune responses are targeted in conjunction with the drug, overall, potentiating leishmanicidal effect and parasite clearance’.

has been amended to:

‘Since both of these approaches are associated with different levels of success, a third approach is to combine the drug with a host-directed therapeutic or therapeutics in a chemoimmunotherapeutic approach, where host immune responses are targeted in conjunction with the drug, overall, potentiating an enhanced leishmanicidal effect and parasite clearance’

We have amended the text with "neglected NTD" (originally Llne 357) so that the text reads:

‘Leishmaniasis is a severely neglected disease despite the immense suffering it places on the host, especially in regions of economic instability. We have improved the quality of the figures and hope they are now suitable’

  1. We have critically analysed the manuscript for problems with the English and had useful feedback from a colleague (Professor Jim Alexander). We have marked the changed text in yellow.  We would like to thank the reviewer for this oversight.

Reviewer 2 Report

The present article represents an intense literature review comparing the available treatment methods against leishmaniasis. As correctly stated by the authors, although leishmaniasis is considered a control priority, it is often overlooked and therefore the topic of the article is of high importance. It should be also noted that there is a lot of missing information for treatment of leishmaniasis and this review article covers a significant proportion in a well organised way. Furthermore, the article is well written and the created figures are nice. 

Although I would recommend publication of the article, I have some corrections, recommendations and suggestions that according to my opinion have to be addressed before acceptance.

Firstly, I find the title misleading. In its current form someone may think that the article concerns all diseases in general. I would recommend to be more specific and make clear that immune responses and drug treatment concern leishmaniasis and not all diseases. 

In lines 57-59 the authors state all Leishmania endemic countries have HIV infected populations. This is not particularly correct. For instance, Greece is an endemic area for leishmaniasis but has no HIV infected populations. Please correct or rephrase

I also suggest to include a section in the Introduction regarding the canine leishmaniasis problem that is also of high importance for some countries. 

Similarly, regarding the treatment methods, it would be more informative to also include the most common treatment methods for dogs, if any, and if there are no particular treatments for those animals, to add a small paragraph regarding the control or therapy of the disease in domestic animals.

Finally, although there are a few things regarding resistance proteins, it would be very interesting to add literature regarding particular Leishmania genotypes that are associated with resistance in any of the treatment methods.

Author Response

We would like to thank the reviewer for his/her kind words and thank them for their time.  In response to the comments we have made the following changes to the manuscript.

  1. Title: we agree with the author and apologise for this oversight.

The original title ‘Can we harness immune responses to improve drug treatment’ amended to ‘Can we harness immune responses to improve drug treatment of leishmaniasis?’

  1. The reviewer said there are no cases of HIV in Greece however, that is not the case as there are articles to show that HIV is present in Greece (1016/j.meegid.2018.04.01010.1007/s10461-019-02402-1); (10.2174/1570162X15666171122165636).

However, to ensure we are factually correct we have amended the test from ‘All Leishmania-endemic countries have HIV infected populations [16] although the total burden of this co-infection is underreported, partly due to remoteness of affected areas’

has been amended to:

‘Therefore, VL may pose a greater heath risk in Leishmania-endemic countries where HIV infected populations are present [18]’

  1. We did not introduce canine leishmaniasis as a topic into this review as we were trying to keep within a 5000-word limit and we wanted to concentrate on the drugs used to treat leishmaniasis and host directed therapies. We have now added the following sentence to the manuscript and directed the reader to suitable literature on this topic.

Inserted text in the introduction (lines 46-48):

‘Reservoir hosts such as dogs, are very important in the transmission of VL in endemic areas, and these hosts should be considered in clinical and veterinary Leishmania control programmes [7,8]’

  1. We have critically analysed the manuscript for problems with the English and had useful feedback from a colleague (Professor Jim Alexander). We have marked the changed text in yellow.  We would like to thank the reviewer for this oversight.